# Vaccine Communication: Appeals and Messengers Most Effective for COVID-19 Vaccine Uptake in Ukraine

**DOI:** 10.3390/vaccines11020279

**Published:** 2023-01-28

**Authors:** Gretchen Schulz, Kristian Balgobin, Alexandra Michel, Rupali J. Limaye

**Affiliations:** 1Department of Population, Family, and Reproductive Health, Johns Hopkins Bloomberg School of Public Health, Baltimore, MD 21205, USA; 2Department of International Health, Johns Hopkins Bloomberg School of Public Health, Baltimore, MD 21205, USA; 3International Vaccine Access Center, Johns Hopkins Bloomberg School of Public Health, Baltimore, MD 21205, USA; 4Department of Epidemiology, Johns Hopkins Bloomberg School of Public Health, Baltimore, MD 21205, USA; 5Department of Health, Behavior & Society, Johns Hopkins Bloomberg School of Public Health, Baltimore, MD 21205, USA

**Keywords:** COVID-19, vaccine communication, vaccine uptake, Ukraine, message appeals

## Abstract

Throughout the COVID-19 pandemic, vaccine communication has been a challenge, particularly as some populations may be highly distrustful of information from public health or government institutions. To better understand the different communication needs in Ukraine, an online survey panel of 168 Ukrainian participants viewed six COVID-19 vaccination ads with three variations on vaccine messaging appeals (potential economic impacts of COVID-19 infection COVID-19 disease outcomes, and social norms related to vaccination) and two different messengers (a peer or a health provider). The ad featuring a health outcome appeal delivered by a healthcare provider was most favored (*n* = 53, 31.6%); however, across demographic categories, including vaccine hesitancy categories, participants expressed high levels of approval for all six variations of the COVID-19 vaccine ads. When participants ranked reasons why someone may not accept the COVID-19 vaccine, the most prevalent beliefs identified were that the vaccine was not safe, and that the vaccine was not effective. Findings from this study suggest that vaccine appeals focused on health outcomes delivered by healthcare providers are preferred by most individuals in Ukraine; however, individuals are motivated by a myriad of factors suggesting that for vaccine messaging to be most effective, communication should be varied in both appeal and messenger.

## 1. Introduction

Vaccine hesitancy, which is the delay or refusal of vaccines despite vaccine availability, has impeded the ability of public health organizations to effectively respond to the COVID-19 pandemic [1]. Hesitancy toward all vaccines has been increasing over the past 15 years and has contributed to a resurgence of vaccine-preventable diseases [2,3]. While there has been an increase in hesitancy toward vaccination globally over the past decade, the underlying reasons for hesitancy vary greatly by context. Misinformation and disinformation spread on social media platforms have further amplified distrust of vaccines, and this distrust has only increased during the COVID-19 pandemic [1,4,5]. Other common reasons for hesitancy are vaccine safety and efficacy concerns [1,4]. In Ukraine, there is a long-standing history of distrust in both the local and national health systems [1,6]. Surveys of Ukrainians suggest that hesitancy toward vaccines can be linked to inaccurate media reporting of vaccine side effects and a high percentage of health care workers being vaccine hesitant [6].

Vaccine coverage rates for childhood vaccines in Ukraine are suboptimal. Between 2017 and 2019, Ukraine experienced a major measles outbreak of more than 100,000 measles infections [6]. A 2019 study of Ukrainian adults found approximately 29% of Ukrainians believed that vaccines generally were safe and only 50% thought vaccines generally were effective [6]. Vaccine hesitancy in Ukraine has been further exacerbated by SARS-CoV-2; at the beginning of the COVID-19 pandemic, surveys found that only between 50–61% of the public were willing to receive the COVID-19 vaccine once available [6]. Ukraine has administered more than 31 million doses and data indicate that less than 35% of the eligible population has received two doses as of February 2022 [7]. COVID-19 vaccine acceptance rates in other European countries are significantly higher, ranging from 48% in Spain to 83% in Denmark and 88% in Ireland [8]. This low rate of vaccination partly stems from distrust due to the lack of transparency of the government as well as low levels of trust in scientists and medical professionals [1,6].

Beginning in February of 2022, the conflict between Ukraine and Russia has further exacerbated the COVID-19 pandemic in Ukraine [9]. During the conflict, vaccine hesitancy sentiments have continued to proliferate on the media platforms that Ukrainians consume, and misinformation surrounding COVID-19 vaccines saturate social media channels [9]. The uncertainty of the conflict has only furthered the amount of misinformation and disinformation surrounding COVID-19 vaccines, leaving many in the country unwilling to accept the vaccine, and the ongoing conflict is likely to further erode the trust in the government and health systems, negatively affecting vaccine uptake [9].

Adequate messaging strategies are critical for the public to adhere to public health recommendations, including vaccine recommendations, as different populations may be receptive to different formats, framing, and styles of communication messaging, particularly populations with high distrust toward health officials and the government including public health sources [10,11,12]. A 2015 literature review of strategies for addressing vaccine hesitancy found that there is no one specific strategy to successfully address vaccine hesitancy [13]. For example, vaccine promotion campaigns in mass media intended for broad, general audiences have the potential to improve attitudes toward vaccine uptake; however, the authors also note that mass media campaigns may be unlikely to appeal to those who are already distrustful or hesitant toward vaccines [13]. As such, for communication to be successful across a range of populations, particularly those who are distrustful of public health authorities, it is critical to employ relevant messengers and messaging that is tailored for the needs of different audiences [13].

Previous research has highlighted the importance for communicators to consider the various components of their communication appeals. A 2021 literature review synthesized vaccine appeals, finding that a focus on social norms, personal narratives, and peer communication may be most effective at changing perceptions about vaccines [14]. The messenger, as well as the message, is also an important feature of communication and can also greatly impact on the effectiveness of communication efforts [15]. Although healthcare professionals have been identified as effective messengers for promoting vaccine uptake, particularly in low-and-middle income countries, not all populations trust health organizations and healthcare professionals when it comes to vaccination [16]. Given the importance of restoring trust in vaccination, this study sought to evaluate three different appeals (health outcomes from COVID-19 infection, vaccination as a social norm, and the economic impact of COVID-19 infection) delivered by two different potential messengers, a healthcare provider compared to a peer, on COVID-19 vaccine acceptance in Ukraine.

## 2. Methods

We conducted an online panel survey of participants in Ukraine. Participants were recruited using SurveyMonkey. Inclusion criteria required all participants to be at least 18 years of age, currently residing in Ukraine, and proficient in written English. To ensure a high level of data quality, we implemented several data quality checks using best practices on online surveys [17,18]. Survey responses were excluded if there were unrealistically short completion times, failure of survey attention checks, and failure to complete the survey including all survey questions.

### 2.1. Survey Items

The survey included general socio-demographic questions such as age (18–24, 25–39, 40–64, 65+), gender (male, female, other, prefer not to say), education level (secondary or high school, bachelor’s degree or 4-year college degree, graduate level degree, or other), as well as pregnancy status (yes, no, not applicable). To ensure usability and clarity, the survey was pretested with 10 participants.

Participants viewed six potential ads for COVID-19 vaccines which were comprised of a specific messenger and specific vaccine message appeal (Appendix A). Ads displayed two individuals engaged in conversation composed of either a healthcare provider speaking with a patient or a peer image depicting two similarly dressed people in conversation. The six ads were randomly shuffled and presented to each participant to reduce order bias. Each of the ads included one of three vaccine appeals: economic benefit, which focused on loss of income and work due to COVID-19 and the protective effect of vaccination against economic loss; health outcome, which focused on the risk of disease presented by COVID-19 and the protective effect of vaccination against disease; and social norms, highlighted how most people in their community have received the COVID-19 vaccine and the protection that vaccination offers for the community. After viewing an ad, participants responded to six questions on their agreement with the ad’s relevance, motivation to obtain a COVID-19 vaccine, motivation to obtain COVID-19 vaccination for their child (if they confirmed being a guardian or parent), if the ad was designed for someone like themselves, and if the ad would prompt the participant to tell a peer or others in their community about the COVID-19 vaccine. Participants were also asked to indicate which of the six ads was most motivating for them to seek out COVID-19 immunization. We collapsed ad preference responses into binary variables (strongly agree/agree vs. strongly disagree/disagree).

We hypothesized that participants categorized as high vaccine hesitancy would have different ad preferences compared to those who were low in vaccine hesitancy, with a preference for peer messengers over health providers. We asked three questions to measure vaccine hesitancy: (1) if they had ever delayed getting a recommended vaccine (2) safety concerns related to COVID-19 vaccines and (3) perception of vaccine effectiveness. We constructed an ordinal scale from 0 to 3 to describe participants’ level of vaccine hesitancy by assigning 1 point for each of these yes/no questions. A cut-off was then used to stratify participants into two groups, lower hesitancy (0–1) and higher hesitancy (2–3).

To further understand any potential differences in messaging preferences we included a ranking question to ascertain specific concerns about COVID-19 vaccines. Participants were asked to rank six reasons why an individual may not want to receive a COVID-19 vaccine. Concerns included safety, vaccine effectiveness, trust in government, vaccine experience, cost, and belief in the existence of COVID-19.

### 2.2. Statistical Analysis

We summarized participant characteristics and responses using the frequency and percentages. We examined the proportion responding positively across constructs for each ad. Variables for some participant characteristics were collapsed due to small numbers in some categories, including age (collapsed to <40 and ≥40 levels for regression models), gender (female and male levels used for regression models as the “prefer not to say” option had only *n* = 3 observations).

We examined participants’ ad preferences by asking participants which ad they most preferred using multivariable multinomial logistic regression. We estimated relative risks and 95% confidence intervals using Ad 1 as the reference group (Health Outcome—Healthcare Provider ad) and a binary variable for vaccine hesitancy as our primary characteristic of interest. We then assessed the relationships between ad characteristics, participant preferences, and participant demographic characteristics in separate multivariable multinomial models. Models included participant socio-demographic data known from the literature to be associated with vaccine attitudes [19,20]. Statistical analysis was performed in Stata 16.1 (Stata Corp, College Station, TX, USA). The study received ethical approval from (blinded for review).

## 3. Results

A total of 395 individuals started the online survey (Figure 1). To ensure that only high-quality participant responses were included, participant responses were excluded if the 55-question survey was completed in <5 min (*n* = 145, 36.7%), attention checks were failed (*n* = 80, 20.3%), or survey responses were incomplete (*n* = 2, 0.5%). Of the total 395 survey responses collected between 4 January 2022 and 5 January 2022, 168 met the inclusion criteria.

Table 1 describes the study participants. The majority of participants were 24–39 years old (*n* = 91, 54.2%), female (*n* = 91, 54.8%), and had a graduate degree (*n* = 70, 41.7%). Fourteen percent of participants reported being currently pregnant (*n* = 24).

Despite more than half of respondents saying they had received a COVID-19 vaccination (*n* = 106, 67.5%), the majority of participants were classified as vaccine hesitant (*n* = 138, 82.1%). Surveys on COVID-19 hesitancy and vaccine uptake in Ukraine have found evidence that Ukraine is “the least vaccinated country in Europe” with only around 40% of the population reporting completion of the initial two doses of COVID-19 vaccine [9]. It is important to note that vaccine uptake behavior is complex and reflects many different contextual factors. Although individuals may express hesitancy, access to a trusted messenger can support vaccine uptake. For example, a UNICEF survey in Ukraine found that a physician’s recommendation on vaccination was highly supportive of vaccine uptake [21]. Across 2000 individuals surveyed in Ukraine, 77% of unvaccinated individuals said they would be vaccinated if recommended by a physician [21]. Of the survey participants categorized as having high vaccine hesitancy, more than half were between the ages of 24–39 (*n* = 91, 54.2%), and had either a bachelor’s (*n* = 58, 34.5%) or graduate degree (*n* = 70, 41.7%) (Table 2).

Participants ranked the most common reasons someone might not want to receive the vaccine. Safety was the main reason why someone may not want to receive the COVID-19 vaccine. This was followed by not believing that vaccines are effective, and then distrust in the government (Figure 2).

Across all six ads, participant level of agreement was high, regardless of message appeal or messenger (level of agreement >90%). The most highly preferred ad included a health outcome messaging appeal paired with a health provider messenger (Table 3). Across all ads, the healthcare provider was the preferred messenger compared to a peer. When asked which of the ads would be the most motivating to obtain a COVID-19 vaccine, 31.5% of participants ranked the ad that paired a health provider with a health outcome message as the ad most likely to motivate them to receive the COVID-19 vaccine. While other findings were not statistically significant, a multinomial logistic regression did illustrate that participants who were 40+ years of age preferred the economic appeal delivered by a peer messenger ad (Appendix A).

## 4. Discussion

Understanding ways to develop influential communication to encourage vaccine uptake is critical. Including an array of vaccine messaging from multiple messengers is key to building greater confidence in the safety, efficacy, and necessity of vaccines [13,14,22]. This study aimed to better understand how different vaccine messaging appeals and messengers may be effectively utilized to combat COVID-19 vaccine hesitancy within Ukraine. Participants largely favored the ads delivered by healthcare providers. The most favored ad of the six was the ad focused on health outcomes delivered by a healthcare provider. This suggests that public health information delivered by healthcare providers in Ukraine is effective at promoting greater vaccine confidence and encouraging greater vaccine uptake. Considering low levels of COVID-19 vaccine uptake paired with high levels of vaccine hesitancy in Ukraine, it is essential that public health outreach adequately tailor vaccine messaging. This is particularly important as evidence on the waning effectiveness of the initial COVID-19 vaccine series demonstrates the need for booster doses to protect against new and emerging virus variants. Continued efforts to support immunization must include effective, evidence-based strategies for developing messaging that is relevant to different populations based on demographic as well as behavioral factors, such as vaccine hesitancy. An important finding from this research is identification of both health providers as well as peers as trusted messengers for COVID-19 vaccination.

A major limitation of this work is that our study population was not nationally representative of Ukraine’s population as we relied upon an online panel. Our survey sample was skewed toward people aged 24–39, and individuals who have obtained at least a college degree. Although there are limitations to generalizing our study population to a representative national population in Ukraine, utilizing online survey tools during a pandemic is an innovative way to gather data while in-person data collection increases risks [23].

Despite these limitations, this is one of the first studies to test different combinations of vaccine appeals and messengers in Ukraine. The results of this study found the health outcome appeal delivered by a healthcare provider was most preferred, and healthcare providers were preferred by most participants compared to a peer messenger. This is an important finding for supporting ongoing COVID-19 vaccination efforts to bring Ukraine’s immunization rate up to the 65–70% vaccine coverage necessary to reach population immunity in the country [6]. Prior research has demonstrated that provider recommendation and discussion with patients about vaccines improves uptake across a variety of vaccines [24,25,26]. Future vaccine communication in Ukraine should use these findings to communicate the importance of COVID-19 vaccination more effectively. We are hopeful that this work will help public health, practitioners, and government agencies to more effectively promote vaccines to a variety of target audiences.

## 5. Conclusions

Vaccine hesitancy continues to be a leading threat to global public health. Finding ways to effectively communicate with a variety of populations and audience is essential to ensuring adequate vaccine uptake in various contexts and is a crucial component to improving vaccine coverage rates and rebuilding trust in public health efforts. The on-going conflict in Ukraine adds yet another layer of difficulty for public health efforts in the region to curb preventable mortality and morbidity due to COVID-19. Our findings are intended to support public-health outreach efforts in Ukraine by testing relevant message components to determine whether particular populations may have distinct preferences. We found evidence that both healthcare providers as well as peers are suitable as trusted messengers for COVID-19 vaccine messaging across different populations, regardless of vaccine hesitancy attitudes, and focusing on the health outcomes of COVID-19 is likely to be a particularly effective strategy for vaccine appeals in Ukraine. Vaccine hesitancy research has demonstrated that even if vaccines are available, safe, and effective, an individual’s motivations to seek out and accept immunization are complex and highly context-dependent. By identifying appropriate trusted messengers and relevant, engaging appeals, we can strengthen the evidence base for effective public communications and support immunization as a critically important public-health intervention.

## Figures and Tables

**Figure 1 vaccines-11-00279-f001:**
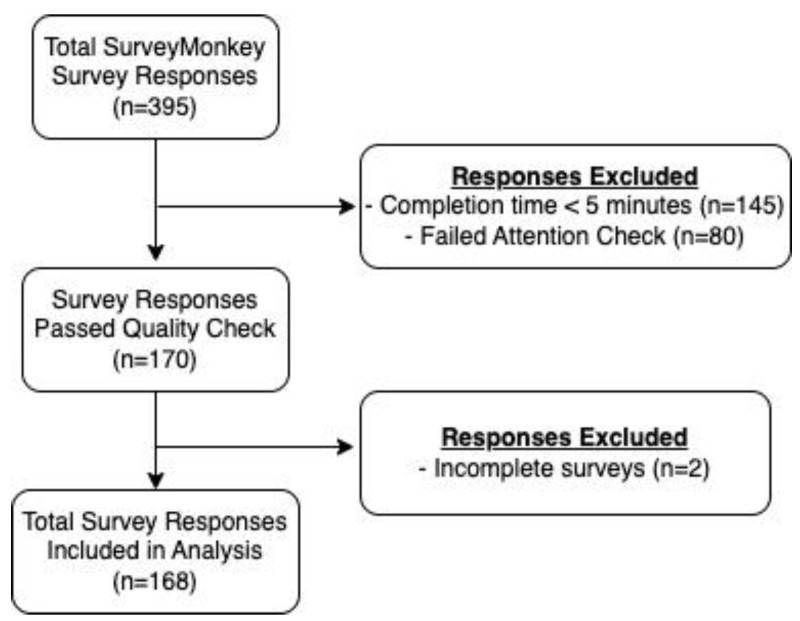
Participant responses flowchart.

**Figure 2 vaccines-11-00279-f002:**
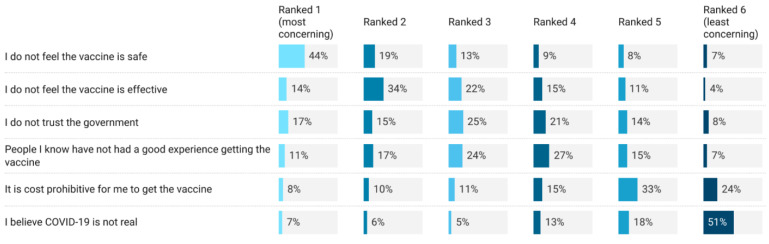
Ranked reasons why someone may not want to receive the COVID-19 vaccine.

**Table 1 vaccines-11-00279-t001:** Participant characteristics and prevalence of vaccine hesitancy (*n* = 168).

Characteristic	No. (%)
**Age**	
18–24	28 (16.7%)
24–39	91 (54.2%)
40–64	45 (26.8%)
65+	4 (2.3%)
**Gender**	
Female	91 (54.8%)
Male	75 (45.2%)
Prefer not to say	2 (1.2%)
**Education**	
Secondary	28 (16.7%)
Bachelor’s degree	58 (34.5%)
Graduate degree	70 (41.7%)
Other	12 (7.1%)
**Currently Pregnant**	
No	113 (67.3%)
Yes	39 (23.2%)
Not applicable	16 (9.5%)
**COVID-19 Vaccinated**	
No	51 (30.4%)
Yes	106 (63.1%)
It is not available in my country	6 (3.5%)
I am not eligible to receive the vaccine	5 (3.0%)
**Vaccine Hesitancy**	
Lower Hesitancy	30 (17.9%)
Higher Hesitancy	138 (82.1%)
**Ever delayed recommended vaccine**	
No	86 (51.2%)
Yes	63 (37.5%)
I do not know	19 (11.3%)
**Concerned COVID-19 vaccine might not prevent the disease**	
Extremely/Moderately	50 (29.8%)
Slightly/Not at all	118 (70.2%)
**Concerned COVID-19 vaccine might not be safe**	
Extremely/Moderately	40 (23.8%)
Slightly/Not at all	128 (76.2%)
**Concerned COVID-19 vaccine might not be safe for pregnant women**	
Extremely/Moderately	81 (48.2%)
Slightly/Not at all	87 (51.8)
**Concerned COVID-19 vaccine might not be safe for children**	
Extremely/Moderately	75 (44.6%)
Slightly/Not at all	93 (55.4%)

**Table 2 vaccines-11-00279-t002:** Participant characteristics and vaccine hesitancy.

Characteristic	Low Hesitancy No. (%)	High Hesitancy No. (%)	Total
Age			
18–24	5 (17.9%)	23 (82.1%)	28 (16.7%)
24–39	14 (15.4%)	77 (84.6%)	91 (54.2%)
40+	11 (22.4%)	38 (77.6%)	49 (29.2%)
Gender			
Female	17 (18.7%)	74 (81.3%)	91 (54.2%)
Male	11 (22.4%)	38 (77.6%)	75 (44.6%)
Prefer not to say	-	-	2 (1.2%)
Education			
Secondary	5 (17.9%)	23 (82.1%)	28 (16.7%)
Bachelor’s degree	8 (13.8%)	50 (86.2%)	58 (34.5%)
Graduate degree	15 (21.4%)	55 (78.6%)	70 (41.7%)
Other	-	-	12 (7.1%)

**Table 3 vaccines-11-00279-t003:** Participant Preferences for Six Possible COVID-19 Vaccine Ads (*n* = 168).

Ad Preference	Frequency (%)
Health Outcome—Healthcare Provider	53 (31.5%)
Health Outcome—Peer	25 (14.9%)
Economic—Healthcare Provider	29 (17.3%)
Economic—Peer	11 (6.6%)
Social Norm—Healthcare Provider	35 (20.8%)
Social Norm—Peer	15 (8.9%)

## Data Availability

Not applicable.

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
