# Peer review of "Vaccine Communication: Appeals and Messengers Most Effective for COVID-19 Vaccine Uptake in Ukraine"

_vaccines, 2023, doi:10.3390/vaccines11020279_

Round 1

Reviewer 1 Report

I congratulate the author for conducting a study in the middle of a conflict. I have some minor comments which I think will improve the paper. 

Methods: 

1. Please mention the order in which the ads were shown to the participants. 

2. The description about collapsing and categorising responses should be moved in the survey items section in line with the description of the particular items. These are not part of the Statistical analysis. 

3. The line about Statistical Software should be moved to the Statistical analysis section. 

4. In 135, please write " we summarized participant characteristics and responses using number and percent" instead of "we summarized participant characteristics and responses". 

Results: 

1: Please check table 1. For example, the percentage for age does not add up to 100%. The percentage for "ever delayed recommended vaccine" - Yes was presented up to two decimal points. 

2. The author should present all the multivariate logistic regressions mentioned in the Statistical Analysis section as a supplementary (citing appropriately in the text) in support of their results. 

Discussion: 

1. I would prefer if the author discussed the key findings (as mentioned in the abstract) in the first paragraph. 

Author Response

Response to Reviewer's Comments/ Concerns

Methods: 

1. Please mention the order in which the ads were shown to the participants. 

I have revised the methods section to explain that the ads were randomly shuffled by the SurveyMonkey survey to reduce order bias, thus the order of the ads were not identical for each participant. 

2. The description about collapsing and categorising responses should be moved in the survey items section in line with the description of the particular items. These are not part of the Statistical analysis. 

As recommended, I moved the sentence about collapsing the ad preference to binary variables to the survey items rather than the statistical analysis section.  

3. The line about Statistical Software should be moved to the Statistical analysis section. 

I have moved the software sentence to the end of the statistical analysis section. 

4. In 135, please write " we summarized participant characteristics and responses using number and percent" instead of "we summarized participant characteristics and responses". 

I have changed this line to reflect that we used frequency and percentages.

Results: 

1: Please check table 1. For example, the percentage for age does not add up to 100%. The percentage for "ever delayed recommended vaccine" - Yes was presented up to two decimal points. 

I have reviewed the data for table 1 and included the "I don't know" and "not applicable" response numbers and percentages. I have also corrected the 2 decimal point error in the table as well. 

2. The author should present all the multivariate logistic regressions mentioned in the Statistical Analysis section as a supplementary (citing appropriately in the text) in support of their results. 

Although these findings were not statistically significant, I have included the multivariable multinomial logistic regression findings in a table in the appendix. 

Discussion: 

  1. I would prefer if the author discussed the key findings (as mentioned in the abstract) in the first paragraph.

I have included information in the discussion about the major findings of this study, including implications of the ad preference shown in the results. 

Reviewer 2 Report

Thanks to submit the manuscript titled as "Vaccine communication: appeals and messengers most effective for COVID-19 vaccine uptake in Ukraine" for the brief report.

Efforts are being made in many countries to increase vaccination coverage, especially for COVID-19 vaccination, and it was interesting to report on the efforts in Ukraine regarding the use of messengers as one of these efforts.

However, there are a lot of technical mistakes in the manuscript. Especially, results of statistical analysis part is insufficient, so the result itself is unclear. Moreover, discussion section is shortage of consideration.  Therefore you should revise this paper substantially.

1. You mentioned about using "multivariable multinomial logistic regression" in the methods section (lines 143). But the details were unclear. Dependent and objective variables for multivariate multinomial logistic regression analysis should each be clearly stated. Normally, when logistic regression analysis is performed, odds ratios for comparison items should be obtained, but this is not stated either. Since this analysis is the most important part of this paper, the methodology should be described in detail.

 2. In the "Results" section, you mentioned about "Despite more than half of respondents saying they had received a COVID-19 vaccination (n=106, 67.5%), the majority of participants were classified as vaccine hesitant (n=138, 82.1%)."(lines 166-168), I found it very strange.  If you hesitated to vaccinate but did vaccinate, I do not see a problem because the result is that you vaccinated regardless of whether you hesitated or not. On the contrary, for example, if, on the contrary, you wanted to vaccinate but did not actually do so, this would be much more problematic. How do you explain this discrepancy?

3. Results of multivariate multinomial logistic regression analysis are insufficiently described. Table 2 explains almost nothing. The main result of a multivariate multinomial logistic regression analysis should be the odds ratio of each objective variable to the dependent variable, and such frequency and percentage notations do not make sense. It is also unclear which items were used as independent variables in the first place, and there is no description of the model itself, so there is no way to evaluate it.

4. Discussion section is also insufficient. You states the limitations of the study, but the causal relationship is unknown because it is a cross-sectional study to begin with. Furthermore, what is the point of conducting this study when more than half of the population has completed vaccination?

The results of the analysis are also inadequate, making it impossible to make a deep argument against the study. 

Reviewer 3 Report

This manuscript briefly reported that the major factors for COVID-19 vaccination appeal and messengers in Ukraine where attracts the most concerns around the world. Through the careful survey and analysis of the investigation outcome, the authors revealed the main needs and factors for effective COVID-19 vaccine communications in Ukraine, which is also meaningful and instructive for vaccinations in other countries. This report would facilitate people understand the situation and attitude of local people toward COVID-19 vaccination in Ukraine, and it is helpful for the prevention of COVID-19 in Ukraine and other areas in the world. Therefore, its worth of publication on Vaccines

Author Response

Thank you for reviewing this manuscript and for your comments.

Reviewer 4 Report

After examining the scientific study, the following considerations may be made. The scientific study is well structured in all its parts. In particular, the premises with which the authors introduced the analysis are clear. Equally clear are the objectives that led the authors to carry out this study and the section on materials and methods. Particular appreciation can also be expressed of the material on which the study was carried out. The data was collected methodically and without bias. The results were consistent and significant and allowed a discussion section full of food for thought. The latter has also been well articulated, in fact presents a first introductory part on in the matter under consideration. The authors then developed a discussion of the results achieved. 

This article is particularly current as it is particularly relevant to the issue of the covid-19 pandemic.

The number and quality of the citations is appropriate, however the scientific relevance of the article could benefit from an expansion of the same. In the specific advice to add the following quotes:

·       Line 36, I suggest to add the following quote: “di Fazio, N., Caporale, M., Fazio, V., Delogu, G., & Frati, P. (2021). Italian law no. 1/2021 on the subject of vaccination against Covid-19 in people with mental disabilities within the nursing homes. La Clinica terapeutica172(5), 414–419. https://doi.org/10.7417/CT.2021.2349

English is well structured in syntax and grammar.

Author Response

Thank you very much for your comments. We have reviewed our citations and have selected those that most support the topic and importance of this manuscript. 

Round 2

Reviewer 2 Report

The revised version, with the reviewers' input, makes much more sense than the first draft. Although the scientific impact is less, I believe that the paper has value as a brief report.